

# Accumulation-based Runoff and Pluvial Flood Estimation Tool

Hannes Leistert[1], Andreas Hänsler[1], Max Schmit[1], Andreas Steinbrich[1], Markus Weiler[1]

[1]Hydrology, Faculty of Environment and Natural Resources, University of Freiburg, Freiburg, Germany

*Correspondence to*: hannes.leistert@hydrology.uni-freiburg.de

**Abstract.** Knowledge about spatially distributed inundation depth and overland flow quantities, as well as related flow velocities, is critical information for establishing a pluvial flood forecasting system and the related disaster management. This kind of information is often derived from computationally demanding simulations with 2-dimensional hydrodynamic models, limiting the number of scenarios for which information can be provided and challenging real-time forecasting. To address this

gap, we developed the model AccRo (**Acc**umulation-based **R**unoff and Fl**o**oding), which is a computationally efficient method to derive maximum inundation depth, maximum flow velocity and maximum specific discharge of a flood event at larger spatial scales, based on an improved flow accumulation method to better represent the spatial extent of inundated areas. To assess the quality of AccRo, we compare the results from the AccRo model with the results of two different state-of-the-art 2-dimensional hydrodynamic models for design cases as well as real-world pluvial flood examples. We find that AccRo is able

to represent both, the analytical solution for the design cases and the simulations of the hydrodynamic models in the real-world example in high quality, well within the range of the two hydrodynamic models. In combination with the low computational requirements, we conclude that AccRo is a valuable tool for assessing pluvial flood hazards.

## 1 Introduction

Pluvial floods have large damage potential as is demonstrated by recent catastrophic events like the pluvial flooding in Texas

in July 2025 or the floods in Spain from October 2024. They are supposed to become more frequent and more intense in the light of global change (e.g. Skougaard Kaspersen et al, 2017). So far, however, warning mechanisms for pluvial floods are rather limited. This is on one hand due to the random nature and fast build-up phase of extreme convective precipitation events (e.g. Li et al, 2021), although probabilistic ensemble forecasts might improve the forecast quality (e.g. Bouttier and Marchal, 2024). On the other hand, there is also the issue of computational resources and adequate lead times connected with the

identification of inundation areas. This is especially important for the real-time warning of pluvial floods in suburban and urban regions, often characterised by a relatively strong role of hydrological processes and relatively large accumulation areas, requiring a more holistic approach to include the impact of runoff generation and hydraulic effects into a warning.

Because this is not yet available, a widely used concept to support pluvial flood risk assessment as well as a baseline for planning emergency management options are so-called pluvial flood maps (e.g. Wimmer and Hovenbitzer, 2025). These maps

map potential inundation areas for certain design rainfall events based on the output of hydrological and/or hydrodynamic



models (Bulti and Abebe, 2020). A step towards issuing a real-time warning of pluvial floods was recently made with the concept for a mesoscale pluvial flood index (PFI) to depict the current hazard of a region being flooded (Weiler et al., 2025). The PFI builds on the spatial extent of flood hazard areas in a certain region. These are defined as areas where the flood poses a danger for people's life. Key variables for most pluvial flood maps as well as the PFI are the maximum inundation depth and

maximum surface flow velocity. The PFI also takes into account the maximum specific discharge. To accurately represent hazard areas, the spatial resolution of the data must be fine enough to represent important flow structures in urban areas (e.g., buildings, streets, underpasses, etc.) as well as in the surrounding areas (farm and forest roads, small creeks, etc.), which frequently deliver water into settlements.

The above-mentioned variables are also a standard output of 2-dimensional (2d) hydrodynamic models. However, especially

when targeting a larger regional scale (e.g. when focusing on sub-urban cases with large accumulation areas of 10-100 km$^2$ or in the case of deriving the PFI) the high computation costs and integration times of 2d hydrodynamic models might become a bottle neck for real-time forecast. Although there are recent developments like GPU based 2d hydrodynamic models (e.g. RIM2D, Apel et al., 2024, or IBER-Plus, Moraru et al., 2023 or SCENARIFY, Buttinger-Kreuzhuber et al., 2022) or the use of ANN for instant forecasting (e.g. Berkhahn et al., 2019), both approaches are bound to huge computation efforts either for

the real time simulation in the case of the GPU models or in the preparation of extensive training data for the ANN models. Especially for mixed regions with larger influence of non-urban areas and diverse hydrological responses, the training of ANN requires a very large dataset (e.g. Reinecke et al, 2024).

On the other hand, there are computationally efficient GIS-based methods available to simulate the flow accumulation in a very reasonable time also for larger areas (Avila-Aceves et al., 2023). However, there are some shortfalls, which are mainly

that the outputs of these methods do not fulfil the data requirements for estimating the pluvial flood maps, since they only provide accumulated flow amounts but no explicit measure for inundation depths or flow velocities. Furthermore, spatial structures of inundated areas extracted from flow accumulation methods are often much more spatially confined than the ones from 2d hydraulic models. Although accumulation-based methods exist that try to mimic the spatial extent of inundated areas depicted in hydraulic models (e.g. FastFlood, van den Bout et al., 2023) they do not yet provide the spatial details needed to

define local flood hazard areas.

In order to obtain a fast estimate of the spatial pattern of maximum water depth, maximum flow velocity and maximum specific discharge at a reasonable spatial resolution allowing us to represent important obstacles as buildings and preferential flow paths (e.g., roads and ditches), we developed the raster-based model AccRo (**Acc**umulation-based **R**unoff and Fl**oo**ding). In our study, we first describe the methodological details of the AccRo and the validation framework we used to compare the

AccRo output with 2d hydrodynamic models, followed by the validation of the results.





## 2 Method

### 2.1 Relating accumulated runoff ($A_s$) to $q$, $w$ and $v$

Central element of AccRo is the flow accumulation function (FAF) of the Python module richDEM (Barnes, 2016). FAF is a very fast method yielding reliable results, for quantifying flow accumulation with a choice of several flow direction approaches

like convergent methods, such as D8 (O'Callaghan and Mark, 1984) or divergent methods, like Quinn method (Quinn et al. 1991). One major advantage in using FAF is that it allows the use of accumulated weights based, e.g., on area or in our case on surface runoff intensities ($s$). Applying FAF with $s$ as a weighting factor yields to spatially distributed estimates of total surface runoff flowing through a raster cell depending on the input $s$ and the flow paths, according to the digital elevation model (DEM). To use FAF to estimate maximum inundation or water depth ($w$), maximum flow velocity ($v$) and maximum

specific discharge ($q$), cumulative surface runoff ($A_s$) must be translated into the appropriate target variables. Furthermore, the accumulation of $s$ in FAF is only controlled by the DEM, ignoring the need to account for changing hydraulic conditions caused by varying water depths. As a result, the second critical element in AccRo is the decoupling of the accumulation from the DEM surface when hydraulic conditions and flow direction change due to varying water depth.

As previously stated, using FAF with $s$ as the weighting factor returns $A_s$. The temporal development of $A_s$ in a specific cell,

however, is not incorporated by FAF. Knowing the travel times is key to estimating $q$, $w$ and $v$, thus we have to include the temporal perspective into AccRo. For a given raster cell, we can distinguish between the time a water parcel needs to flow though the cell ($t_c$), the forward time i.e. the time the water parcel needs to reach another cell (typically the outlet), and the backward flow time ($t_b$), which is the duration of flow in a single cell caused by delayed inflow from cells further uphill (see Fig. 1). In general, $t_b$ will be shorter on top of hills or convex structures (less cells accumulated) and larger in valleys and

reaches the maximum value at the catchment outlet. The concepts for $t_b$ and $t_f$ are widely used in hydrology, e.g. $t_b$ for water age calculations (Benettin et al. 2015) and pollutant transport (Zhang et al. 2023) and $t_f$ for isochrones and hydrographs estimations (Olivera et al. 1999, Saghafian et al. 2002).





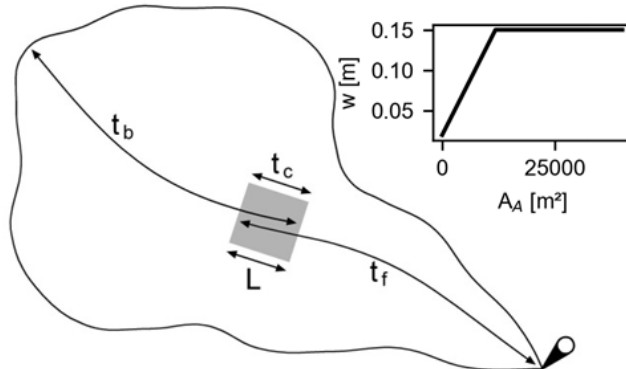


**Figure 1: Scheme of the definitions of relevant times a water parcel travels in a catchment and the assumed initial relationship between water depth and area weighted flow accumulation.**

### 2.1.1 Spatially distributed estimation of backward flow time ($t_b$)

For a given water depth, the flow velocity ($v_c$) and hence the flow time ($t_c$) through a raster cell can be derived from the slope and hydraulic radius based on the Gauckler-Manning-Strickler (GMS) equation:

$$v_c = k \times R^{\frac{2}{3}} \times i^{0.5} \qquad \text{Equation 1}$$

$$t_c = \frac{l}{v_c} \qquad \text{Equation 2}$$

with

$k$ – Strickler surface roughness [$L^{1/3}$/T]

$R$ – hydraulic radius [L] – in our case water depth in a cell

$i$ – slope [L/L]

$l$ – cell length [L]


The slope is derived from the DEM and the roughness values can be estimated from spatially explicit landuse or surface cover information (e.g. LUBW, 2016). Adding up $t_c$ along a flow path from the top to the bottom (top-down approach) yields the backward flow time ($t_b$) for each cell along this flow path, which represents the duration of flow in a cell caused by flow times from upslope cells finally draining through the given cell (see Fig. 1 for schematic). Since our method is designed for events

with rather high $s$ values (pluvial flood events), we initially estimate $t_c$ (and hence also $t_b$) only for water depths between 2 cm and 15 cm. To assign an initial water depth ($w_c$) for the calculation of $t_c$, we developed a simple linear relationship (see Eq. 3) between the $w_c$ and area weighted flow accumulation ($A_A$) derived from FAF using the Quinn Method, with an intercept equal



to 2 cm ($w_{sf}$) and a slope of $1\times10^{-5}$ m$^{-1}$ (see inlet in Fig. 1). The reason for using the Quinn Method for area and runoff weighted

accumulation is that it is a divergent method representing flow processes, which is more realistic than convergent methods on

hillslopes and in valleys. Flow occurs to all downslope neighbours proportional to $i$ and to the tangent of the angles. $w_{sf}$ is the

water depth under which we assume sheet flow (HLNUG, 2020). The slope represents the quotient of a water depth ($w_r$) above

which we assume that roughness is independent of changes in water depth (i.e. larger than 10 cm) and a critical accumulation

area of 10,000 m² ($A_{Acrit}$). For $A_{Acrit}$ we consider that the entire area would generate runoff, the resulting water depth reaches

values of 10 cm ($w_r$) or more due to the accumulation. As previously stated, the maximum is set to 15 cm. The use of this

linear equation yields only catchment or area-specific results that are unaffected by a specific runoff event.

$$w_c = \begin{cases} 0.02m & w_c < 0.02m \\ \frac{w_r}{A_{crit}} \times A_A + w_{sf} & 0.02m \leq w_c \leq 0.15m \\ 0.15m & 0.15m < w_c \end{cases} \qquad \text{Equation 3}$$

Having defined the cell-specific $t_c$ we now have to identify for each cell all upslope cells eventually draining into the cell of

interest to calculate the backward flow time $t_b$. For the top-down approach (starting with the highest DEM value), we use a

sink-filled DEM (modified with the richDEM epsilon filling module, Barnes, 2014) as a baseline for an area-weighted D8

accumulation. We use the convergent D8 method of FAF here, since we just want to select the upslope cell with the largest

inflow and thus the most relevant $t_b$ value. $t_b$ in a cell is then the sum of $t_c$ in this cell and the relevant $t_b$ value upslope. To find

the most relevant upward $t_b$ value of the neighbouring eight cells, we queried four criteria to select the relevant $t_b$ of the eight

neighbouring cells:

(1) DEM value is higher,

(2) D8 accumulation is lower,

(3) of all neighbours that fulfil criteria (1) and (2) D8 accumulation is maximal

(4) of all neighbours that fulfil criteria (1) and (2) $t_b$ is maximal.

Since not all 4 criteria are always fulfilled, the importance decreases from (1) to (4). If criterion (2) is not fulfilled, a D8

accumulation weighted $t_b$ of all neighbours fulfilling (1) is used as a replacement. If even criterion (1) is not fulfilled, criteria

(2) and (3) are applied to neighbours with equal DEM values. Figure 2 exemplarily shows for a small catchment (ca. 3 km²;

see section 2.4 and Fig. 6 for more details) the spatial distribution of the variables used to calculate $t_b$ (panels a to c) as well as

the resulting $t_b$ (Fig. 2d).



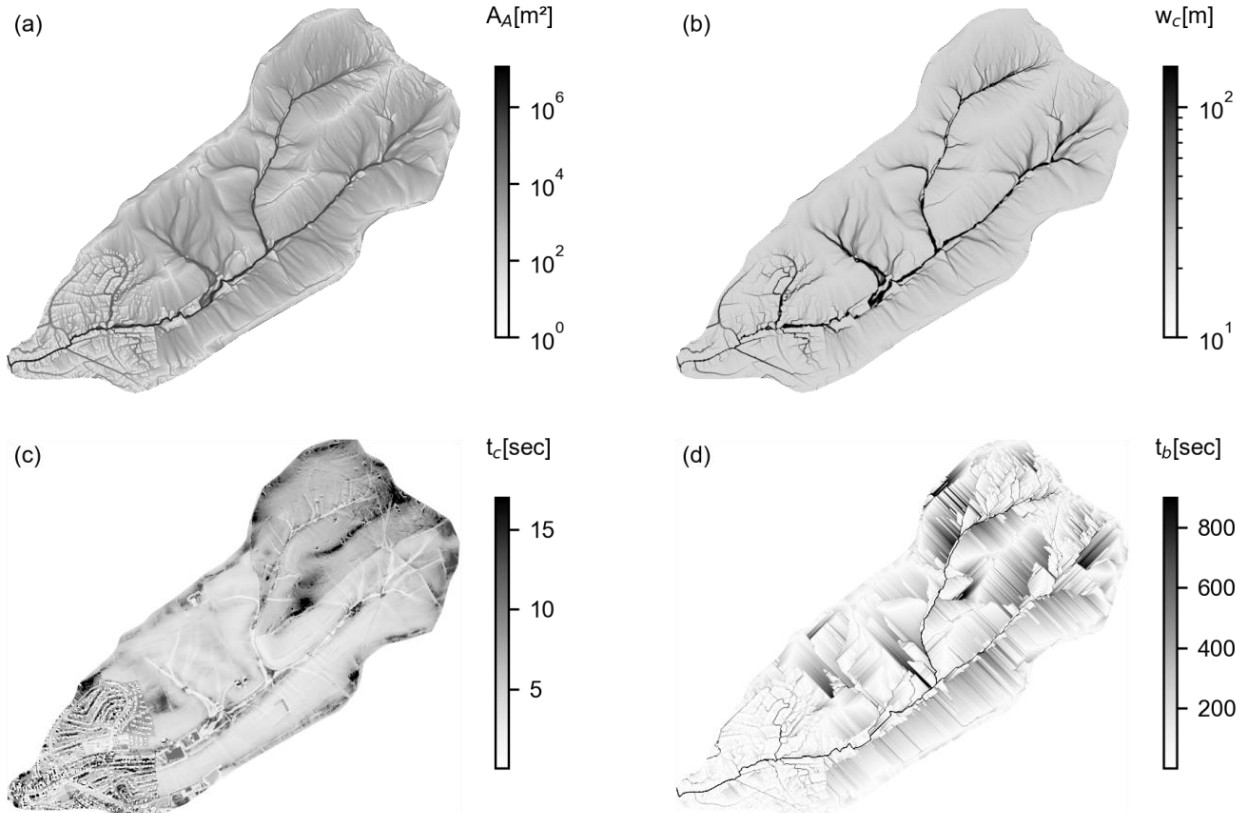

**Figure 2: Example of spatially distributed variables to calculate backward flow time in a small catchment (see section 2.4). (a) Area flow accumulation (m²) → (b) Area specific water depth (m) → (c) cell flow time (sec) → (d) backward flow time (sec)**

### 2.1.2 From surface runoff accumulation to cell-specific maximum discharge using backward flow time

In the previous section, we showed how to calculate for each raster cell an event-independent backward flow time $t_b$. However, in order to determine the maximum specific discharge $q_{max}$ at a particular cell, we must connect this data to the event-specific $s$ response and the accumulated surface runoff $A_s$. In essence, the following principle guides the estimation of $q_{max}$:

$$q_{max} = A_s \times l \times F \ [\text{L}^3/(\text{T*L})] \qquad \text{Equation 4}$$

respectively surface discharge per cell

$$Q_{max} = A_s \times l^2 \times F \ [\text{L}^3/\text{T}] \qquad \text{Equation 5}$$





where $F$ [1/T] represents the factor to transfer $A_s$ [L] to $q_{max}$ [L³/T] and $l$ [L] the cell size.

The next section provides additional details concerning the estimation of $F$ (see Fig. 3 for an example). We start with the most

basic scenario: a raster cell with no inflows from neighbouring cells, such as the top of the hill at the watershed boundary. In

this case, the temporal response of generated runoff $s$ of the specified cell (see Fig. 3a) is equal to the response of surface

runoff. The maximum $s$ is equivalent to the maximum surface discharge. In this case, $t_b$ equals $t_c$, with $t_c$ getting closer to zero

as the cell size decreases (i.e., it is 0 for a point). In this scenario, $F$ is the quotient of the cell-maximum value of $s$ ($s_{max}$) and

the product of its discrete time unit ($\Delta t$) and its sum ($\sum s$) (see Eq. 6).

$F = \frac{s_{max}}{\sum s \times \Delta t}$          Equation 6

However, the situation is different for cells that receive inflow from neighbouring cells. In this case, $t_b$ is greater than $t_c$, and

we obtain distinct overlapping $s$ responses. The degree of overlap depends on the chosen time step of $s$ (since $s$ is an intensity).

As a result, $s_{max}$ and $\sum s$ change for cells with $t_b > t_c$ (Fig 3b). A particular cell's $\sum s$ (and $A_s$) will increase with its $t_b$, which

essentially indicates that more cells will eventually inflow into the cell. For cells with larger $t_b$, $F$ gradually decreases because

$s_{max}$ reaches a maximum at a specific $t_b$ and will not increase further due to the delayed runoff from the inflow cells. (Fig. 3c)

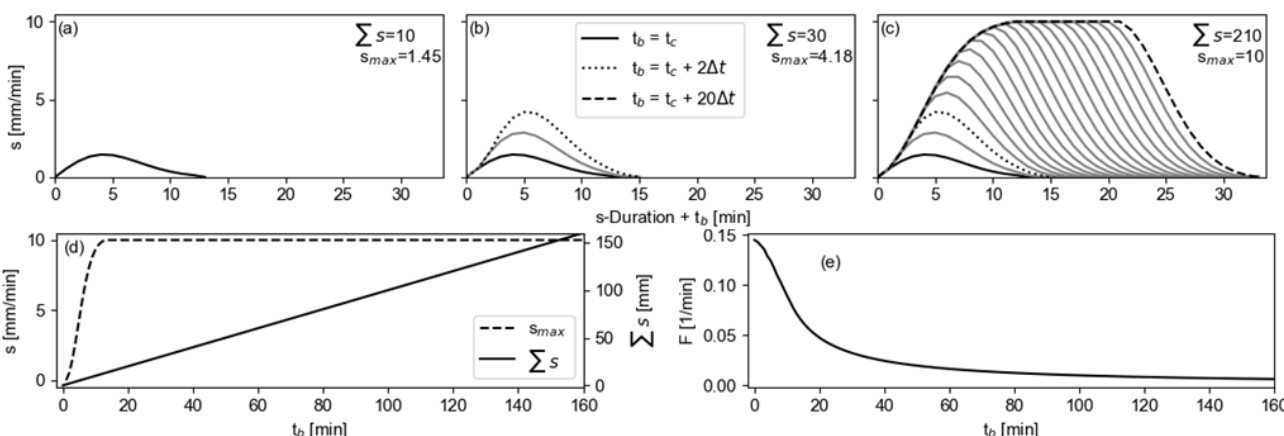

**Figure 3: Example of how $F$ is developing in dependence of $s$ and $t_b$**


To visualise the evolution of $s_{max}$, $\sum s$ and $F$ with increasing $t_b$ in Fig 3., we defined an exemplary flow path. In Fig. 3a, $t_b$

equals $t_c$, $s_{max} = 1.45$ mm/min and $\sum s = 10mm$ and so $F = 0.145$/min. In Fig. 3b-c we increase $t_b$ (2 minutes and 20 minutes)

and the corresponding $s$ response. The shift of overlap is defined by the underlying time step of the intensity. So in Fig. 3b, $t_b$

equals $t_c +2$ min, $s_{max} = 4.18$ mm/min and $\sum s = 30mm$ and so $F = 0.14$/min and in Fig. 3c, $t_b$ equals $t_c +20$ min, $s_{max} =$

$10$ mm/min and $\sum s = 210mm$ and so $F = 0.048$/min. Calculating this for all possible $t_b$ in a catchment, we receive a



continuous curve for $s_{max}$, $\sum s$ (Fig. 3d) and $F$ (Fig. 3e) in dependence of $t_b$. Based on this explanation, it is evident that $F$ does not depend on the time unit of $s$. However, the $F$ factor changes if the quantity and/or duration of the response varies.

As the spatial and temporal variability of $s$ is relevant for deriving the $F$ curve, the response of $s$ should represent an average response over a specific area. This is usually not fulfilled anymore if the area is becoming too large since the variability in

individual $s$ responses will significantly vary due to different spatial factors influencing the runoff generation, such as vegetation, soil type, moisture, and precipitation. Hence, we target for a size of around 2x2 km² for which a specific response of F is valid. This area can be e.g. a small catchment, some part of a catchment or the area covered by a grid cell from radar-based precipitation input.

### 2.1.3 From maximum specific discharge to maximum water inundation to maximum flow velocity

With the assumption that $q$, $w$ and $v$ are related to each other, it's now possible to calculate w and v based on q using Gauckler-Manning-Strickler equation for each cell (see Eq. 1). Since the hydraulic radius ($R$) is not known, we assume that the cross-sectional width is large compared to water depth, and so we set $R$ equal to $w$. On hillslopes this assumption is mostly given, but in channels this assumption might lead to higher velocities.

By rearranging Eq. 1 we get


$$Q = v \times C = k \times R^{\frac{2}{3}} \times i^{0.5} \times C \qquad \text{Equation 7}$$

and with $R = w$ and

$$C = w \times l \qquad \text{Equation 8}$$

we get

$$w = \left( \frac{\frac{Q}{1000}}{i^{0.5} \times k \times l} \right)^{3/5} \qquad \text{Equation 9}$$

with

$C$ – cross-sectional area of flow [L²]

$l$ – cell size [L]

### 2.2 Representation of changing hydraulic conditions due to inundation

As already introduced before, the use of FAF typically results in water accumulation, either adhering to the steepest descent
in the D8 approach or encompassing all downward gradients for diverging methods, resulting in relatively small inundated





areas. Hydraulic models, on the other hand, attempt to simulate reality by allowing flowing water to initially fill structures such as channels, hollows and sinks before eventually overflowing banks or other structures, resulting in different flow pathways due to the adapting water depths as compared to the initial DEM structures. As a result, the flow accumulation analysis should take into account shifting hydraulic conditions in order to mimic more natural inundation processes.

We use spatially distributed surface runoff ($s$) generated from a hydrological model as a weighting factor in the FAF. The quantity and location of computed accumulated runoff ($A_s$) are determined by the flow direction defined by the DEM and the spatial pattern of $s$. To allow for the expansion of flow pathways in AccRo to mimic the inundation processes, the accumulation is done iteratively by partitioning the total amount of $s$ into $n$ equal fractions ($s_{frac}$). Utilising the previously established methodology for calculating $q$ (Eq. 4) from the generated $A_s$, we can calculate $w$ for each cell based on $q$ after each iteration.

Adding $w$ after each iteration to the elevation before the iteration, we can produce a modified DEM surface, with certain regions already inundated. This adjusted DEM is then the new reference for the FAF in the next iteration step. By repeating this procedure for multiple iteration steps, the accumulated surface runoff generates larger inundated areas and levels out the cross sections (see schematic example in Fig. 4), in contrast to calculating $A_s$ only once without modifying the DEM (Fig. 4 right column).


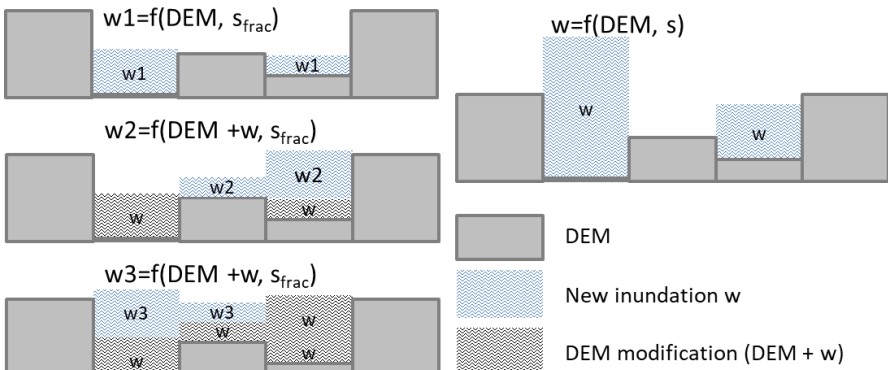

**Figure 4: Schematic illustration of the iterative process. Left: $s_{frac}$ is accumulated in three iterations, altering the DEM with the resulting inundation and therefore changing the accumulation location of each iteration. Right: $s$ is accumulated once without**
**altering the DEM.**

Fig. 5 depicts the spatial effect of the iterative flow accumulation procedure for the example of a small test catchment (see Fig. 6 for details). The maps show the maximum inundation after 72 iterations (left) and the setup where $s$ is accumulated only
once (right). Note that the sum of $s$ is the same for both examples. Narrow structures become more extended structures and additional regions become flooded with more iterations. We additionally included the water depth at a specific cross section





(small inlet figures) to highlight the iteration setup and the resulting inundation, whereas in the case without iteration an unrealistic water hill builds up.

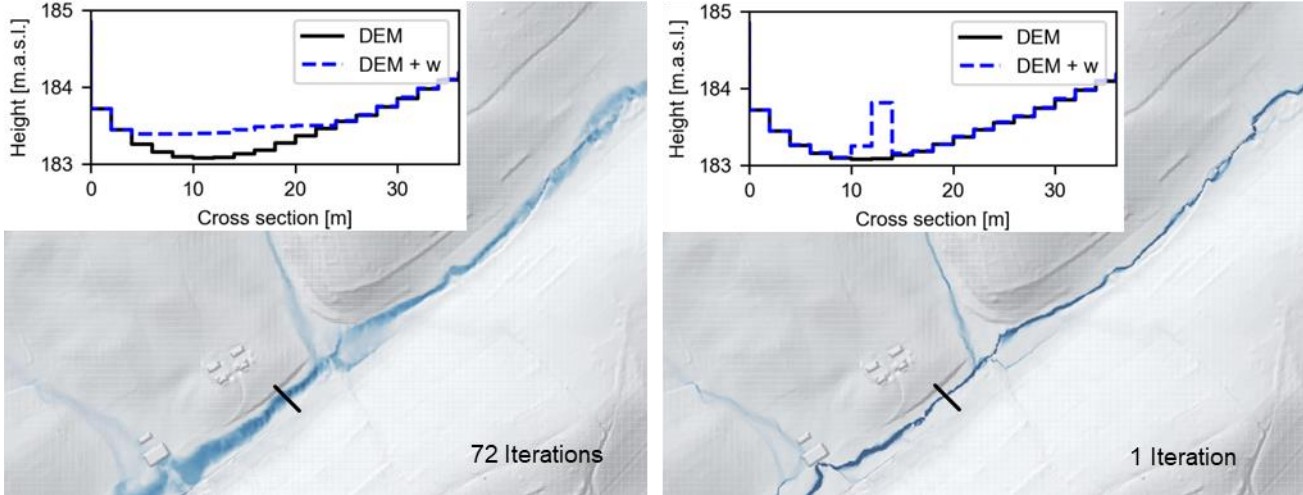


**Figure 5: Effects of the iterative approach. Left: inundation in meter when 72 iterations are applied, right inundation when one iteration is applied. The sum of *s* is the same for both simulations. The black line in the centre represents the location of the cross section.**

While $q$ and $w$ are calculated for each iteration step and then added up to determine the respective maximum values, $v$ is not calculated throughout the iteration process. As a result, only one computation of $v$ is performed after the iteration process using Eq. 1. To account for changes in slopes caused by inundated areas, we recalculate $i$ by incorporating $w$, which means that the slope is now determined by the gradient of the water surface (DEM plus inundation depth).

### 2.2.1 Sinks

For handling sinks, we have developed two approaches. Sinks may be excluded by pre-filling them prior to the iteration, or they may be included and subsequently filled during the iterative process. For the latter, the locations and volumes of sinks are calculated by subtracting the non-filled from the filled DEM. To avoid redistributing of a too large amount of $s$ inside a sink, only $A_s$ of cells representing the local DEM minima in sinks for the relevant iteration step are considered. Redistribution of water within sinks is done by an internal sink iteration routine. This routine redistributes the water within the sinks to level out

DEM plus water depth, ensuring that no cells outside of the sink are affected. Surplus water, or inundation that exceeds the sink depth, is added to $s_{frac}$ in the subsequent iteration step.



### 2.2.2 Defining the number of iterations

As shown in Fig. 4 and Fig. 5, the spatial expansion of the inundation area potentially increases with the number of iterations. The optimal number of iterations, however, is not a fixed value but depends on catchment and event characteristics. In order to produce realistic results in rather flat regions, more iteration steps are needed than in regions with greater topographical variation. Furthermore, higher runoff intensities require more iteration steps than lower intensities. To consider these aspects, AccRo initially estimates the portion of total runoff that would generate a maximum inundation of 10 cm in an accumulation area of 10 km² at one iteration. For this purpose, a single initial run with only one iteration of normalised total runoff $s/\max(s)$ is computed. To calculate the number of iterations $n$, the resulting (spatial) maximum inundation depth is divided by 0.1 m and multiplied by the maximum total runoff as shown in Eq. 9.

$$n = \frac{max(w)}{0.1} \times max(s) \qquad\qquad \text{Equation 10}$$

We evaluated this approach for different regions and various cases with different runoff intensities. We discovered that when n exceeds 100, the findings become quite identical. As a result, we set $n_{max}$ to 120 and $n_{min}$ to 12, allowing for some spatial expansion under low runoff amounts.

### 2.3 Estimation of flood hydrographs

As a by-product of our approach in AccRo, we can also derive the flood hydrograph at defined locations. We use the idea of the geomorphological instantaneous unit hydrograph (GIUH) (e.g., Rigon et al. 2016) and the estimated maximum flow velocity as well as the water volume stored in sinks to calculate the flood hydrograph. For this, we use $v$ to calculate the forward flow time ($t_f$) to a given location in the catchment (in general the outlet, but can be any) with the SAGA module 'ta_hydrology" (Maximum Flow Path Length) (Conrad et al. 2015). The amount of water retained in sinks is used to reduce the response of $s$ i.e. input into GIUH. As $v$ is the maximum velocity of the underlying event, one can expect that the derived flood hydrograph is faster compared to a hydrograph of a 2d hydrodynamic model.

### 2.4 Validation framework

Since pluvial floods usually inundate areas outside streams and channels, measured data of flooding extent or flood hydrographs for different rainfall runoff events are rare. Instead of using real observations, we evaluate AccRo with model simulations of two hydrodynamic models: HydroAs version 6.2.2 (Hydrotec, 2025) and RIM2D (Apel et al. 2024, Version Jan 2025). To evaluate the quality of AccRo results, we defined different scenarios, for which we conducted simulations with the three different models. First, we simulated artificially created hillslopes under steady-state situations (Fig. 6 a, b), where analytical solutions can be derived. Second, we simulated pluvial flooding in a real-world catchment (Fig. 6c).



For the artificial situations, we simulated overland flow on a planar hillside (1000 by 2000 m) with a constant slope of 0.03 m/m (see Fig. 6a). For the second situation a channel of 2 m width is located between two declining surfaces (with a constant slope of 0.03 m/m) with an overall slope along the channel axis of 0.03 m/m. In both cases the cell size is 1x1 m², and a constant input of $s$ was only applied along the uppermost rows (see blue lines in Fig. 6 a and b). For the channel experiment, the total input was set to 1 m³/s but distributed equally along the upper row. When steady state is reached, this value should be found in the channel. For the hillslope experiment, $s$ was set to a constant value of 40 l/s per cell for a width of 500 m on the top of the hillslope. The simulation time was long enough so that steady-state conditions could be reached. In both cases, a constant roughness over the entire domain was assumed; however, we repeated the simulation using different roughness estimates.

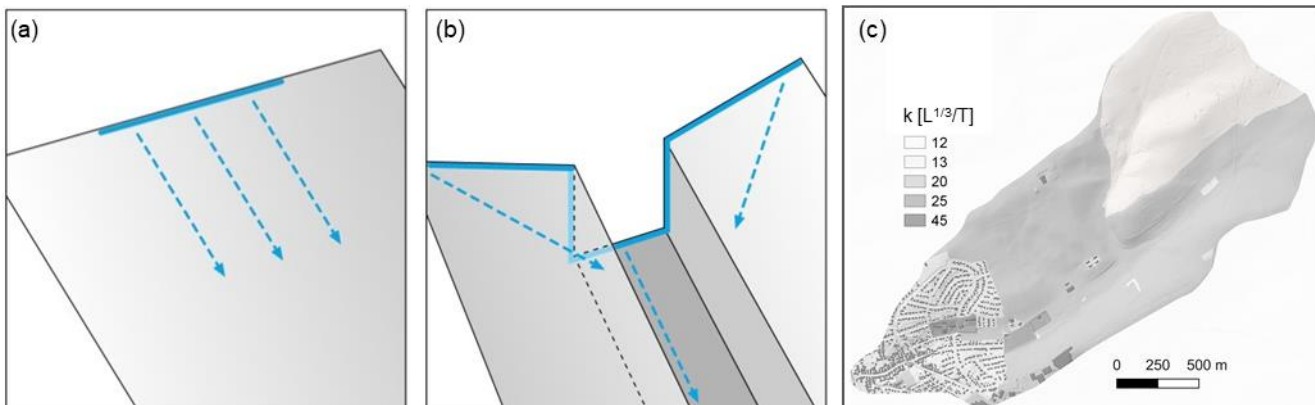

**Figure 6: Overview of the setups used in the validation framework. The two panels on the left depict the design setup with a hillslope (a) and channel (b) case. The blue lines on the top represent the location of constant $s$ input, the arrows indicate predominant flow directions. The real-world catchment Riedgraben is depicted on the right-hand side (c). Here the colours indicate the Strickler roughness values.**

For the real-world scenario, we used the catchment Riedgraben, which is located in southwestern Germany. The catchment area is approximately 3 km², of which 0.67 km² is urban (Bretten-Diedelsheim) and the rest is divided into agricultural and forest areas. The assigned roughness values based on the land use are shown in Fig. 6c. The hydrological input $s$ was calculated with the model RoGeR (Steinbrich, 2016) for an observed extreme rainfall event as well as for three design events. These were defined based on the 1h extreme rainfall event with a return period of 30 and 100 years and defined probable maximum precipitation PMP for this area (LUBW, 2016) (Tab. 1). The spatial and temporal distributed runoff intensities were then used as input into the three different models. The spatial resolution for RIM2D and AccRo was set to 2x2 m². The DEM and hydraulic roughness were obtained from the TIN data used by HydroAs. HydroAs results were rasterized with the HydrAS





tool ha2raster (Hydrotec, 2025) so that a one-to-one comparison of the three models was possible. The focus of the evaluation was on the spatial pattern of maximum water depth, maximum specific discharge and maximum flow velocity.


**Table 1: Summary of selected rainfall and runoff event characteristics.**

| Event | Duration (min) | Precipitation (mm) | Total runoff (mm) |
|---|---|---|---|
| moderate (T=30 years) | 60 | 28.5 | 3.4 |
| heavy (T=100 years) | 60 | 45.7 | 10.0 |
| extreme (PMP) | 60 | 117.6 | 67.0 |
| observed (measured P) | 135 | 67.7 | 19.5 |

## 3. Results

### 3.1 Steady-state simulations

Tables 2 and 3 show a comparison of the Gauckler-Manning-Strickler (GMS) solutions and the three model results. To avoid
boundary effects, the values were taken at the centre of the row 10 metres above the end of the hillslope or channel. For the steady-state situation of the hillslope scenario (Tab. 2), AccRo and HydroAs match the expected values, independent of the roughness conditions. RIM2D results were only stable for the high roughness values, and lateral dispersion seems to be rather large in RIM2D (see Fig. 7a). This and the fact that some water is already lost through the left/right boundaries before reaching the bottom of the slope leads to a lower water depth in the middle of the hillslope. Therefore, RIM2D underestimated $w_{max}$,
$q_{max}$ and $v_{max}$.

**Table 2: Model results of the hillslope scenario; specific discharge at steady state is 40 l/(s*m). $w$ and $v$ are calculated with Gauckler-Manning-Strickler with $i$ =0.03, $k$ = 50, 26.5, 10 m$^{1/3}$/s and $q$ =40 l/s (cross-sectional flow width >> water depth). Unstable simulations are identified with n.s.***

| | Roughness [m$^{1/3}$/s] | Calculated with GMS | AccRo | RIM2D | HydroAs |
|---|---|---|---|---|---|
| | 50 | 0.04 | 0.04 | n.s.* | 0.04 |
| $w_{max}$ [m] | 26.5 | 0.06 | 0.06 | n.s.* | 0.06 |
| | 10 | 0.1 | 0.1 | 0.08 | 0.1 |
| | 50 | 40 | 40 | n.s.* | 40 |
| $q_{max}$ [l/(s*m)] | 26.5 | 40 | 40 | n.s.* | 40 |
| | 10 | 40 | 40 | 25 | 40 |
| | 50 | 1.0 | 1.0 | n.s.* | 1.0 |
| $v_{max}$ [m/s] | 26.5 | 0.69 | 0.69 | n.s.* | 0.67 |
| | 10 | 0.38 | 0.38 | 0.17 | 0.39 |




For the channel scenario (Tab. 3), HydroAS results are slightly closer to the expected values than the values of AccRo for roughness values of 10 and 26.5 $m^{1/3}$/s. Here water depths of AccRo are less than expected. As mentioned before, AccRo calculates $w_{max}$ and $v_{max}$ without any knowledge of a structure like channel width (i.e. 2 m or 2 cells). Instead of using the 'real' hydraulic radius, it uses water depth (see section 2.1.2). The specific discharge of 0.5 m³/(s*m) (1 m³/s distributed in a 2 m

wide channel) is equal to the GMS results. RIM2D results are again only stable for the high roughness values. $w_{max}$ of RIM2D is equal to AccRo, but $v_{max}$ of RIM2D is lower than $v_{max}$ of the other two models.

Note that for RIM2D the relation $q = w * v$ is not fulfilled under stationary conditions, since v in RIM2D is a secondary variable only, while q and w are numerically calculated.


**Table 3: Model results of the designed channel situation; specific discharge at steady state is 1 m³/(s*m). w and v are calculated with GMS equation with $i =0.03$, $k = 50, 26.5, 10$ $m^{1/3}$/s and $q =1$ m³/s (cross-sectional flow width = channel width = 2 m). Cell size is 1x1 m². Unstable simulations are identified with n.s.***

|  | Roughness [$m^{1/3}$/s] | Calculated with GMS | AccRo | RIM2D | HydroAs |
|---|---|---|---|---|---|
| $w_{max}$ [m] | 50 | 0.194 | 0.18 | n.s.* | n.s.* |
|  | 26.5 | 0.293 | 0.26 | n.s.* | 0.287 |
|  | 10 | 0.57 | 0.47 | 0.47 | 0.502 |
| $q_{max}$ [l/(s*m)] | 50 | 500 | 500 | n.s.* | n.s.* |
|  | 26.5 | 500 | 500 | n.s.* | 500 |
|  | 10 | 500 | 500 | 500 | 500 |
| $v_{max}$ [m/s] | 50 | 2.578 | 2.77 | n.s.* | n.s.* |
|  | 26.5 | 1.707 | 1.89 | n.s.* | 1.745 |
|  | 10 | 0.88 | 1.05 | 0.65 | 0.995 |

Figure 7a compares the cross-sectional values of the inundation depth along the hillslope case for a roughness of 10 $m^{1/3}$/s. The inundation depth in the middle of the hillslope (x = 500 m) is equal to the values in Tab. 2 for HydroAs and AccRo.

In any case, comparing the input's dispersion downslope reveals that the 500 m wide rectangular input spreads along the flow path. The inundation depth of HydroAs and AccRo remains equal in the middle of the hillslope, and the spatial extension downward is comparable. After 1500 m, the inundated area extended on both sides by approximately 100 m for HydroAs and

120 m for AccRo. The RIM2D dispersion is much more pronounced, and after 1500 m flow distance, the inundated area reaches the limits of the hillslope. Actually, RIM2D hits the left and right boundaries after around 1040 m, but the other models do not reach them until 2000 m. Figure 7b depicts the cross section of the channel after 1500 metres. As shown in Tab. 3, the water depths of AccRo and RIM2D are equal, hence AccRo's profile is below HydroAS.




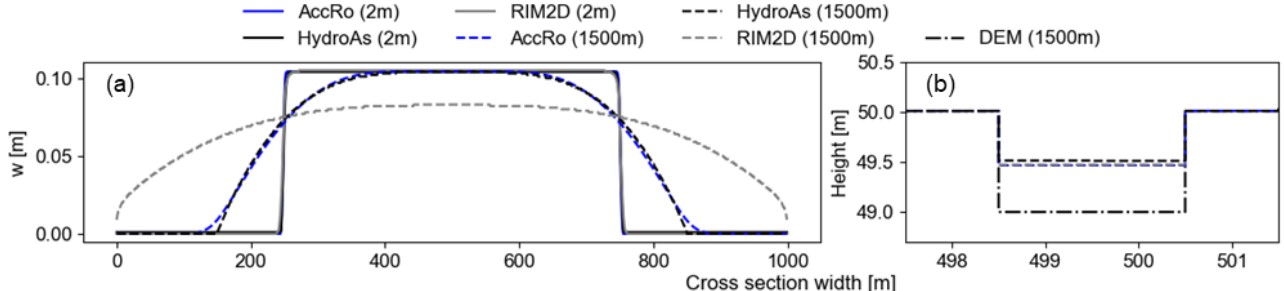

**Figure 7: Cross section of inundation depths at top of the hillslope (2 m) and after 1500 m of the hillslope scenario under stationary conditions (a). Cross section of the channel scenario at 1500 m (b) showing the DEM plus inundation depth.**

## 3.2 Catchment simulations

The simulations of the real catchment differ from the scenarios described in section 3.1 due to temporal and spatial non-stationary input and due to a much more complex hydraulic situation. We evaluate and compare the three target values $w_{max}$, $q_{max}$ and $v_{max}$ in three ways: their distributions to evaluate the overall probabilities of exceedance without spatial explicit comparisons, visually through maps and direct grid-cell to grid-cell comparison, including the slope of a linear regression and the correlation coefficient. In addition, we compare the hydrographs of the hydraulic models to those generated using the GIUH approach.





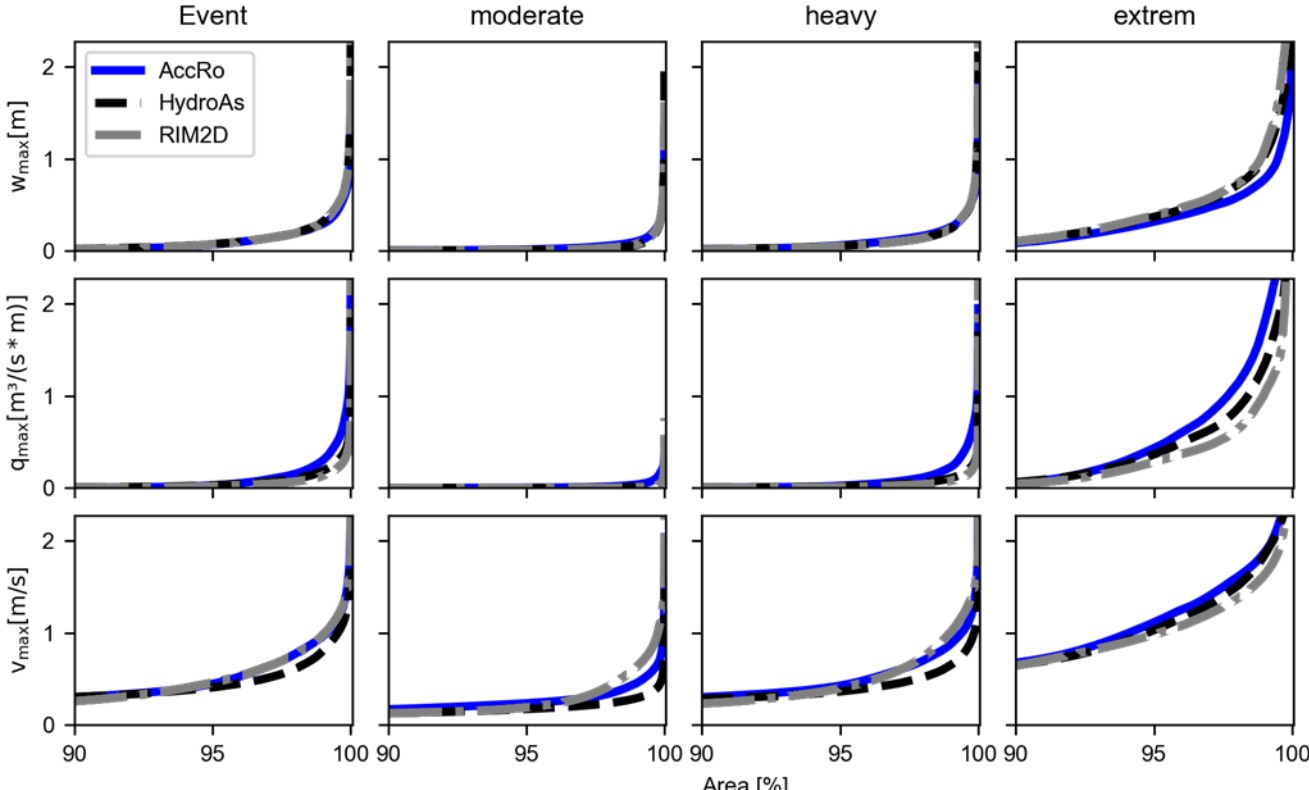

**Figure 8: Comparison of the exceedance probability distribution of $w_{max}$, $q_{max}$ and $v_{max}$ for the four different $s$**

In Fig. 8 the exceedance probability distributions of catchment area with $w_{max}$, $q_{max}$ and $v_{max}$ (rows) are shown for the 4 different events (columns). To highlight the most relevant inundation area, only the upper 10 percent is shown. For all variables and cases the 3 models show rather similar results. Only for the extreme design event (PMP) does AccRo have slightly higher $q_{max}$ and $v_{max}$ values and slightly lower $w_{max}$ values than the other two hydraulic models. The high agreement between the three models' results is also visible in the spatial patterns of inundation depth for the observed event (see Fig. 9 and Supplementary

material Figs. S1 to S3 for other parameters and cases), where all three methods provide extremely similar spatial patterns of maximum inundation depths, particularly for the observed event.



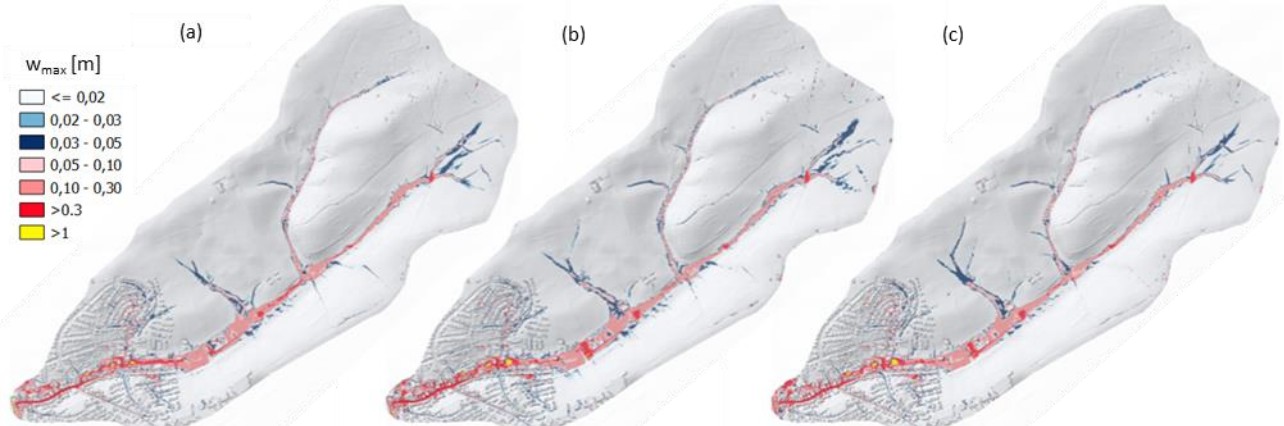

**Figure 9: Visual comparison of $w_{max}$ for $s$ input 'event' as simulated by AccRo (a), HydroAs (b) and RIM2D (c)**


To assess a stricter metric, we examined the simulation results of the three models for the parameters $w_{max}$, $q_{max}$ and $v_{max}$ for all four cases on a grid-cell by grid-cell basis (see Fig. 10 for the observed event and the supplemental materials for the three other scenarios). For $w_{max}$ (Fig. 10, upper row), AccRo matches the two hydraulic models very well, with the vast majority of cells plotting close to the one-to-one line and only a slight tendency for underestimating the inundation depth. For $q_{max}$ and

$v_{max}$, AccRo generally simulates larger values than HydroAs and RIM2D, which is most pronounced in the case of $q_{max}$ (Fig. 10, central row). However, the correlation between the two hydraulic models is not as strong for $q_{max}$ and $v_{max}$ as it is for $w_{max}$. Especially for $v_{max}$, RIM2D appears to have significantly slower maximum flow velocities than HydroAs (Fig. 10, bottom row), which is in the same order of magnitude as the differences between AccRo and the hydraulic models. If we apply the same analysis for the 3 scenarios (see Supplementary material), we find the same patterns as for the observed event case. In

general, AccRo and HydroAs show a better comparison than these two models with RIM2D.





**Figure 10: Scatterplot of the three models. First row $w_{max}$ (m), second row $q_{max}$ (m³/(m*s) and third row $v_{max}$ (m/s) for the real event.**
**In addition, the linear regression (grey) and 1:1 line (dashed black) as well as the slope of the linear regression (m) and the Pearson correlation coefficient (r) are shown.**

Figure 11 shows the hydrographs for the four events. Compared to the hydraulic models, the hydrographs at the catchment outlet of AccRo show an earlier rise and faster drop. The overall response and total discharge are always lower for hydraulic





models, most likely because the simulations of the hydraulic models are stopped before the catchment is completely dry and they also typically retain some water in the catchment as the flow velocities decrease for very low inundation depths. Under more extreme conditions (event and extreme case), the three models produce more similar hydrograph responses.

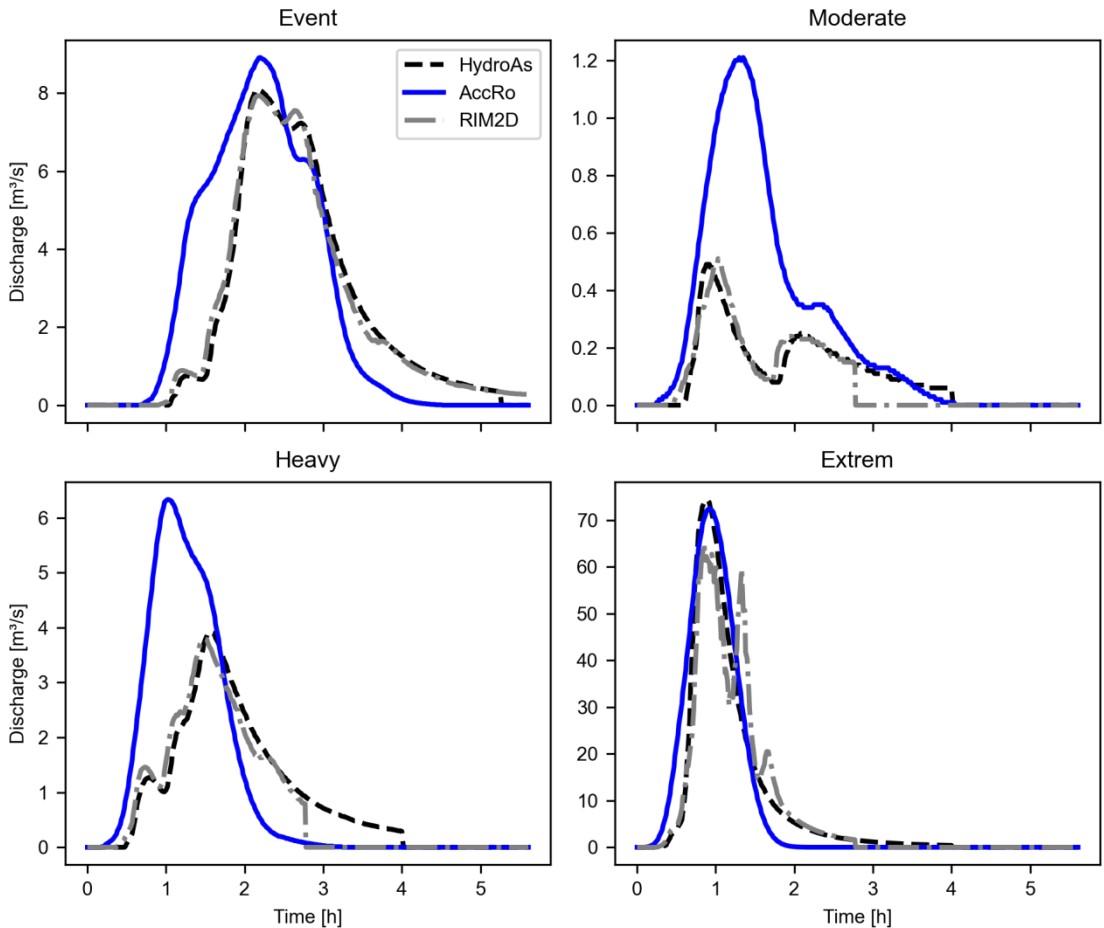


**Figure 11: Hydrographs at the catchment outlet for the four simulated events**

## 4. Discussion

The validation experiment shows that for the design cases, AccRo is able to reproduce the 'hard facts' of the analytical
solutions. For the real world case, however, hard facts are not available, since there is neither a database of systematically observed inundation areas available nor analytical solutions producible. Hence, we use the output of two state-of-the art





hydrodynamic models as reference cases, acknowledging that differences in the outcome of the different methods could be attributed to both shortcomings in AccRo and shortcomings in the hydrodynamic models. Hence, we do not aim for a perfect match of AccRo and the hydrodynamic models, but rather at a 'realistic' outcome of AccRo.

Given the results for the event and scenario simulations for the Riedgraben catchment we see that AccRo tends to have higher $q$ amounts then the hydrodynamic models. In AccRo this value depends on the one hand on the accumulated surface runoff as well as the empirically derived function $F$. $F$ represents a combination of area specific (DEM and roughness) and event (surface runoff response) specific characteristics. The latter takes account of the fact that for the same precipitation amounts, different areas generate different runoff but also that the same area can react differently due to different preconditions, such as

moisture conditions. Investigating the behaviour of $F$ for different types of events we find that $F$ indeed is varying quite substantially (not shown), however it becomes evident that after a specific $t_b$ (twice the duration of the event), the values for $F$ are converging.

In the case of the two hydrodynamic models, maximum specific discharge is calculated as the temporal maximum of the product of $w$ and $v$ at each time step, which means that it accumulates potential errors in both variables and therefore might be

not so robust than the individual parameters. Given the fact that $w$ is rather similar in all three models, also enhances the confidence in the $q_{max}$ output of AccRo, since in AccRo $w_{max}$ is directly calculated from $q_{max}$. Differences in $v_{max}$ are mainly obvious when compared to RIM2D and for the case of the hydrograph. While $v_{max}$ in RIM2D is known to be generally to slow, due to the inclusion of a numerical diffusion term to enhance model stability (Bates et al., 2010), the fast buildup and decrease of the hydrograph peak in AccRo is immanent, since max $w_{max}$ is used to calculate forward flow time ($t_f$) for each raster cell

through the entire runoff event.

Like every model AccRo is based on a series of assumptions and empirical approaches, not all to be justifiable by hand. Examples for these are e.g. the approach to estimate the ideal number of iterations or exact size and definition of the area $F$ is representative for. A major advantage of AccRo, when compared to the hydrodynamic models, however, is the numerical

stability of the approach since it is independent of a model's internal time step (see the design test cases, where both hydrodynamic models had stability issues). This is also beneficial when it comes to computational efficiency. On top of the number of grid boxes, computation time in AccRo mainly depends on the number of iterations, while in hydrodynamic models the duration of the event – and more importantly – the adaptation of the model's internal time step to the horizontal resolution or to stability criteria is required. For the test case of the Riedgraben (~0.75Mio grid boxes), AccRo and RIM2D both simulated

the event in less than 2 minutes (real time) what is fast if compared to HydroAs simulation time of ca 2 hours. However, AccRo and HydroAs were running on standard CPUs, whereas RIM was iterated on a high-efficiency GPU (NVIDIA H100). The downside of this is that AccRo only can model the maximum states of $w$, $q$, and $v$ and not the temporal evolution of these variables.

In the light of using AccRo as a tool for the generation of local pluvial flood maps or real-time inundation forecasting, it

certainly would be beneficial to include processes into the model so far not tackled. Especially the representation of the capacity



of culverts or road passages would be aspects that should be tackled when further developing AccRo. In this turn, also focusing on computational efficiency and parallelization would increase the suitability of AccRo for forecasting even further.

## 5. Conclusion

In conclusion, this paper describes an alternative method to hydraulic models to derive critical variables $w_{max}$, $q_{max}$ and $v_{max}$
caused by pluvial flooding. The comparison with analytical solutions and hydraulic models shows good agreement (also for other events and test cases not included in this study). Given this finding and keeping in mind its computational efficiency demonstrates the suitability of AccRo for operational use cases. Still AccRo has some limitations. On top to still missing processes, AccRo cannot show the temporal development of the variables. So a meaningful application of AccRo depends on the issue. With the use of the GIUH method, the temporal aspect can be included in the simulation, but the assumption of a
constant velocity for one cell over the entire runoff event is not realistic. So here again, if the focus of the simulation is mainly the height of peak discharge, it might be justified. Otherwise a hydraulic simulation should be preferred.





# 6. Appendices

**Table A1: List of variables**

| Variable name | Description |
| --- | --- |
| $A_A$ | Area weighted Flow Accumulation (L²) |
| $A_{Acrit}$ | Constant (critical accumulation = 10.000 m²) |
| $A_s$ | Runoff weighted Flow Accumulation (L) |
| $C$ | Cross sectional area of flow [L²] |
| DEM | Digital Elevation Model [L] |
| $F$ | Factor to transfer $A_s$ to q [1/T] |
| $i$ | Slope [L/L] |
| $k$ | Strickler coefficient for determining surface roughness [L$^{1/3}$/T] |
| $l$ | Cell size [L] |
| $L, T$ | Dimensions Length, Time |
| $p$ | Precipitation rate [L/T] |
| $Q$ | Discharge [L³/T] |
| $q$ | Specific discharge [L³/(T*L)] |
| $R$ | Hydraulic radius [L] |
| $s$ | Surface runoff intensity [L/T] |
| $s_{max}, w_{max}, q_{max}, v_{max}$ | Represents maximum value of parameters |
| $S_r$ | Sinks [L] |
| $t_b$ | Backward flow time [T]. Overall duration of flow through a cell caused by transit times from upslope cells [T] |
| $t_c$ | Flow time of a cell based on flow velocity [T] |
| $v$ | Flow velocity [L/T] |
| $w$ | Inundation depth [L] |
| $w_r$ | Water depth above roughness is independent of w [L] |
| $w_{sf}$ | Upper limit of water depth for sheet flow [L] |





**7. Code and data availability**

The manuscript describes the latest version of AccRo. This version of the model is available at Zenodo under https://doi.org/10.5281/zenodo.17153807 (Leistert, 2025), as are input data and scripts to run the model. All primary data and scripts to produce the plots for all the simulations presented in this paper are also available at Zenodo under https://doi.org/10.5281/zenodo.17154005 (Leistert et al., 2025).

**8. Author contribution**

HL: Method and model code development, Simulations, Data analyzation, Visualization. AH: Method development, RIM2D Simulations. AS: preparation of GIS data. MS: Method development, preparation of GIS data. MW: Supervision, Method development, Conceptualization. All authors contributed to the manuscript

**9. Competing interests**

The authors declare that they have no conflict of interest.

**10. Financial support**

This work was partly conducted within the AVOSS project (funded by the Federal Ministry of Research, Technology and Space in the frame of the WaX program) as well as within the research activities on heavy rainfall at the Chair of Hydrology, University of Freiburg (funded by the State Office for the Environment, Measurements and Nature Conservation of the Federal
State of Baden-Württemberg (LUBW) and the Regierungspräsidium (governing council) Stuttgart).

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
