# Peer review of "Accumulation-based Runoff and Pluvial Flood Estimation Tool"

_EGUsphere, 2025_

## Author Comment (AC1)

**Authors' responses to comments of Reviewer #1:**

We appreciate your review and comments on our manuscript, "*Accumulation-based Runoff and Pluvial Flood Estimation Tool*". Your feedback is valuable to us, and we will make the recommended revisions accordingly. We provide detailed responses to each of your comments below.

In the manuscript titled "*Accumulation-based Runoff and Pluvial Flood Estimation Tool*", the authors present a novel and improved raster-based model to represent key hydrodynamic variables such as maximum inundation depth, maximum flow velocity and maximum specific discharge. The model results are compared against those from two 2D models. The raster-based model achieves a level of accuracy comparable to both 2D models while significantly reducing computational cost. The topic is relevant and timely, and the manuscript is generally clear and well-structured. Overall, I consider this work an interesting and useful contribution to flood modeling and a potential tool for real-time forecasting. For these reasons, I recommend the manuscript for publication in GMD after minor revision.

**General comments**:

**Lines 39-47**: Several 2D models implemented on GPUs can achieve faster-than-real-time performance even for large computational domains, with efficiency sufficient for real-time forecasting. However, this performance strongly depends on the characteristics of the simulated case. The main limitation arises when large domains include localized features (e.g., gullies, small rills) that require fine spatial resolution, leading to small time steps and many operations. I suggest clarifying these situations in the manuscript, as the proposed raster-based model could represent a valuable alternative under such conditions.

Thank you for pointing this out. We will add a brief section on this highlighting the fact that this can have a substantial impact on the performance of 2D hydrodynamic models.

**Lines 56-60**: I suggest emphasizing the novelty of the work in this paragraph. Highlighting how this approach differs from existing methods would help readers better understand the main contribution of the study.

Thank you for pointing this out. We will elaborate a bit more on the novelty of AccRo already here.

**Section "*Introduction*"**: I recommend adding a short paragraph at the end of the *Introduction* to briefly outline the structure of the manuscript, summarizing the content of each section.

We will add a short paragraph like: 'The Paper is structured as follows. In section 2 we describe the methodological details of the AccRo as well as the validation framework we used to verify AccRo output in comparison to 2d-hydraulic models. In section 3 the results of the validation are presented. Discussion of the findings and suggestions for possible further improvements are provided in section 4. The paper ends with a conclusion section.'

**Figure 6**: The figure illustrates the details of the test cases. However, I recommend including a more detailed representation of cases (a) and (b), indicating relevant dimensions such as length and width. This would improve the clarity of the test case setup and facilitate reproducibility.

We will add more details regarding dimensions etc. in Figure 6.

**Table 2**: The table presents results from both 2D models and the raster-based model. However, two of the three simulations using RIM2D are unstable, and the remaining simulation produces results that differ substantially from the others. It is unclear how the model can be unstable for an analytical case. The authors should consider either using alternative software, modifying the simulated case, or removing the RIM2D column entirely, as the results are not informative when the model is unstable, and in the stable configuration, one of the reference 2D models provides results significantly different from the other models.

Actually we think that already the information that RIM2D and HydroAs are in some cases not stable, whereas AccRo is, might already be a result. However, we will emphasize this in more detail in the text. In addition, we will get in touch with the developers of RIM2D in order to find a configuration which might be a bit more stable. Since we include both models for comparison in the real world case as well we wanted to have the same setup in the design case to be consistent o our analysis pathway.

**Section "*Discussion*"**: The authors compare model results using the Figures 7, 8, 9, 11, etc., but the analysis is entirely qualitative. I recommend including at least one quantitative performance metric (e.g., Mean Absolute Error (MAE) or Root-Mean-Square Error (RMSE)) to provide a clearer comparison between models. It is not necessary to compute these metrics for all variables, but, for example, MAE, Peak Percentage Difference, or Peak Time Difference could be reported for the discharges in Figure 11. These metrics would strengthen the discussion.

Thank you for pointing this out. We will add some more quantitative performance metrics for our results presented, as suggested.

**Section "*Discussion*"**: I suggest adding a table summarizing the computational cost for each model and test case. This would make the discussion of computational efficiency clearer and more concise.

Very good suggestion. We will add a table with computation times and computation systems

The abstract states that "*… AccRo is a valuable tool for assessing pluvial flood hazards.*" but the conclusions note limitations regarding temporal development and assumptions of constant velocity. I suggest revising the last sentence of the abstract to better reflect these limitations.

Thank you for pointing this out. We will revise the sentence.

**Specific comments:**

**Lines 10-16**: I recommend homogenizing verb tenses for consistency. For example, the authors alternate between "*we developed…*" and "*we find…*" within the same paragraph.

We will homogenize verb tenses

**Figure 1**: The variable $L$ is included but not defined. If it represents cell length, a uniform notation should be used, as the same variable is defined as $l$ in the text.

L is the dimension length and l is the cell size. The definition of both parameters is provided in Table A1: list of variables. We will make sure that this becomes clearer and once more check the text for uniform notation of variables.

**Equation 3**: Values "$0.02m$" and "$0.15m$" should include a space between the number and the unit, and the unit should not be italicized, consistent with the formatting used elsewhere in the manuscript: "0.02 m" and "0.15 m".

We will include spaces and change the format.

**Figure 3**: Values for $\sum s$ are included but units are not specified. The same applies to $s_{max}$.

We will specify the units

**Line 446**: The authors state "*… (also for other events and test cases not included in this study).*" I recommend either including these additional results in an Appendix, together with the corresponding references (if any), or removing this sentence.

We wanted to show that on top of the analysis framework presented, we meanwhile got a lot of data and test case where we compare AccRo with 2d-Models. However, since we do not always have the setup with the 3 models we did not want to include the detailed results in the manuscript in order to keep the consistency of the evaluation framework presented here. Hence, we will remove this sentence.

**Technical corrections**:

**Line 7**: Replace "*with two-dimensional hydrodynamic models, …*" with "*with 2-dimensional hydrodynamic models, …*".

Will be changed accordingly.

**Line 13**: Replace "*state-of-the-art two-dimensional*" with "*state-of-the-art 2-dimensional*".

Will be changed accordingly.

**Line 47**: Replace "*Reinecke et al., 2024*" with "*Reinecke et al, 2024*".

Will be changed accordingly.

**Line 164**: Include a space between $\sum s$ and "*change*".

Will be changed accordingly.

**Figure 8**: Add a period at the end of the figure caption.

Will be changed accordingly.

---

## Author Comment (AC2)

**Authors' responses to comments of Reviewer #2:**

*We appreciate your review and comments on our manuscript, "Accumulation-based Runoff and Pluvial Flood Estimation Tool". Your feedback is valuable to us, and we will make the recommended revisions accordingly. We provide detailed responses to each of your comments below.*

The manuscript presents AccRo, a new raster-based model for rapid pluvial flood estimation. AccRo computes peak flood variables, maximum inundation depth, flow velocity, and specific discharge, using an improved flow accumulation method combined with Manning's equation. The authors validate AccRo against two 2D hydrodynamic models (HydroAS and RIM2D) for idealized hillslopes and a real catchment, finding that AccRo reproduces the reference models' results closely (generally within the spread of the two 2D models) while being computationally efficient (sub-minute runtimes on CPU vs hours for a 2D model). This work is timely and significant for real-time flood forecasting, offering a novel balance between speed and spatial detail by bridging purely empirical GIS methods and heavy hydrodynamic simulations.

**Major Comments**

1. The introduction would benefit from a clearer discussion of when traditional 2D models (even GPU-accelerated) struggle, to justify AccRo's niche. While the authors note that 2D hydrodynamic models can be computationally prohibitive at large scales, they should acknowledge that some GPU-based models achieve near real-time speeds for certain cases. The key limitation is when fine spatial resolution is needed for localized features (gullies, small channels) in large domains, forcing tiny time-steps and high computational cost. Emphasizing this context will help readers understand why a tool like AccRo is valuable under those conditions. Additionally, the novelty of AccRo compared to existing fast flood tools (e.g., FastFlood by van den Bout et al., 2023) should be highlighted explicitly. What does AccRo do differently (e.g., iterative DEM updating to expand inundation extent, inclusion of flow velocity outputs, etc.) that marks a meaningful improvement? A brief sentence or paragraph at the end of the Introduction outlining the unique contributions of AccRo.

   *Thank you for pointing this out. This suggestion is actually in line with what Reviewer 1 suggested. We will make clearer that the performance of 2D hydrodynamic models can substantially be weekend by finer spatial resolution needed in order to represent localized features.*

   *Furthermore, we will provide more information with respect to the novelty of AccRo compared to existing fast floods tool.*

2. The evaluation relies on surrogate modeling (comparing AccRo to other models) due to scarce pluvial flood observations. While this approach is reasonable, it should be stated more plainly as a limitation. The manuscript should discuss the implications of having no directly observed flood extents or depths for validation. In particular, the authors could acknowledge that agreement with two models (HydroAS and RIM2D) demonstrates consistency, but not absolute "ground truth" accuracy. If any observed data are available, incorporating them would strengthen the validation. If not, the discussion might stress that future work should seek opportunities to test AccRo against measured flood outcomes to further build confidence in its predictive skill.

   *You are certainly right. Unfortunately, in the case of pluvial flood events, 'real' observations are sparse. This is why we designed the two folded evaluation setup –*

once for the design cases where we can estimate the results analytically, and in the other case for the real event. The latter case is by the way the official benchmark case for engineering companies in order to submit an offer for pluvial risk assessments in the federal state of Baden-Württemberg - but even for this case there is no real observation data available.

We will follow your suggestions and state more clearly that the evaluation with observed data would be much more valuable for the evaluation. And we will discuss in more detail that consistency with hydrodynamic models doesn't imply a ground truth accuracy.

3. It should be made clearer in the Abstract and Conclusions that AccRo provides *static* peak flood outputs, not a full dynamic simulation. The manuscript's conclusion does note that AccRo cannot capture the temporal development of flooding and assumes a constant peak velocity per cell, which is "not realistic" over an event's duration. These are important caveats. The Abstract currently touts AccRo as a "valuable tool for assessing pluvial flood hazards" without mentioning its static nature; meanwhile, the Conclusions temper this by recommending a full 2D model if temporal details are needed. To avoid misinterpretation, the authors should adjust the Abstract's final sentence to reflect that AccRo is most useful for rapid hazard mapping of peak conditions, whereas it does not simulate flood timing or dynamics. This balanced wording will ensure readers (and practitioners) understand the model's scope.

Again you are certainly right. We will adapt the respective sections to avoid misinterpretation.

**Minor Comments**

1. **Line 39–47:** Consider inserting a phrase to clarify that GPU-accelerated 2D models *can* run faster-than-real-time in some cases, but face limitations at high resolution. This will nuance the statement about computational bottlenecks for 2D models.

Thank you for your suggestion. We will include a brief section accordingly.

2. **Figure 1:** The variable **L** is shown but not defined in the caption or text. If L represents cell length (which is denoted as *l* elsewhere), use a consistent symbol to avoid confusion.

L is the dimension length and l is the cell size. The definition of both parameters is provided in Table A1: list of variables. We will make sure that this becomes clearer and once more check the text for uniform notation of variables.

3. **Table 2:** Since RIM2D had unstable runs for two scenarios, consider adding a footnote or notation (e.g., "n.s." as used) to explicitly indicate "simulation did not stabilize" for those entries. Also, it may help to gray out or italicize those values to show they are not valid results. This avoids misinterpretation of blank or zero values.

Thank you for this suggestion. We will change it accordingly. Furthermore, we will get in touch with the developers of RIM2D in order to find a configuration which might be a bit more stable.

4. **Line 445–446:** The sentence referencing "other events and test cases not included in this study" should be revised. Either provide those additional results in an Appendix/Supplement (with appropriate references) or remove the claim. As is, it hints at evidence that the reader cannot evaluate.

   We wanted to show that on top of the analysis framework presented, we meanwhile got a lot of data and test case where we compare AccRo with 2d-models.

   However, since we do not always have the setup with the 3 models we did not want to include the detailed results in the manuscript in order to keep the consistency of the evaluation framework presented here.

   Hence, we will remove this sentence.

**Technical Edits:**

**Line 21:** *Citation format.* Ensure proper punctuation in citations with "et al." – for example, "Skougaard Kaspersen et al., 2017" (include a comma *and* period) instead of "et al, 2017". This occurs in a few places; please check all references for consistency with journal style.

Thank you for pointing this out. We will check all references for consistency with journal style.

**Figure 8 caption:** Add a period at the end of the caption for completeness (it currently seems to lack ending punctuation).

Yes, will be accordingly

**Line 20:** *Wording.* The phrase "They are supposed to become more frequent…" could be revised for formality. Consider "expected to become more frequent and more intense…" for a more scientific tone.

Thank you for pointing this out. We will change it, so it reads:  …. *They are expected to become frequent and more intense in the light of global change ...*

**Lines 29:** *Wording clarity.* "These maps map potential inundation areas…" is repetitive. Consider rephrasing to "These maps depict potential inundation areas…" to avoid using "map" as both noun and verb in the same sentence.

Thank you for pointing this out. We will change it, so it reads:  … *maps depict potential inundation areas …*

**Line 385:** *Wording.* "AccRo and HydroAs show a better comparison than… with RIM2D." – This phrasing is a bit awkward. It would be clearer to say "closer agreement" (e.g., "AccRo and HydroAS agree more closely with each other than either does with RIM2D"). This emphasizes the intended meaning that those two models produce more similar results.

Thank you for pointing this out. We will change it, so it reads:  *::: AccRo and HydroAs agree more closely with each other than …*

**Line 447:** *Grammar.* "On top to still missing processes, AccRo cannot show the temporal development…" is grammatically unclear. I would suggest: "In addition to the processes still missing, AccRo cannot show the temporal development of the variables." This rephrase makes the sentence structure clearer.

Thank you for pointing this out. We will change it, as suggested